# Does the Introduction of Alien Species Represent a Sanitary Threat for Native Species? The Case of the Eastern Cottontail *Sylvilagus floridanus* in Italy

**DOI:** 10.3390/life10080142

**Published:** 2020-08-06

**Authors:** Paolo Tizzani, Daniela Andrade, Anna Rita Molinar Min, Andrea Peano, Pier Giuseppe Meneguz

**Affiliations:** Department of Veterinary Sciences, University of Turin, Largo Paolo Braccini 2, 10095 Grugliasco, Italy; daniela_andrade@msn.com (D.A.); annarita.molinar@unito.it (A.R.M.M.); andrea.peano@unito.it (A.P.); piergiuseppe.meneguz@unito.it (P.G.M.)

**Keywords:** *Sylvilagus floridanus*, alien parasites, invasive species, sanitary risk

## Abstract

Introduction of alien species is a well-known threat to biodiversity. Where newly introduced, alien species may pose a risk for the local ecological community by competing for resources or by introducing pathogens. *Sylvilagus floridanus* is an American lagomorph introduced into Europe in the second half of 20th century, for hunting. This study evaluated the structure and epidemiological characteristics of the gastrointestinal parasite community in an introduced population of *S. floridanus* in the Province of Alessandria (Piedmont Region—Italy). Three alien parasites were reported out of 271 animals: *Obeliscoides cuniculi* in the stomach, *Trichostrongylus calcaratus* in the small intestine, and *Passarulus nonnanulatus* in the large intestine. All these nematodes are commonly reported in *S. floridanus* in its natural range, but they represent alien species in Europe. The report of these alien parasites is an example of the unexpected consequences caused by the introduction of non-native vertebrates. The documented introduction of new pathogens may alter the parasite community of the native lagomorphs, with possible long-term effects on local ecological dynamics.

## 1. Introduction

More than 16,000 alien species are established outside their native range at a global level [1]. Around one thousand have documented ecological effects, and the taxonomic groups causing the highest impacts are terrestrial invertebrates and terrestrial plants [2]. Alien species might represent a threat to biodiversity, leading to the extinction of indigenous species. They are harmful in different ways: by competing for resources, inter-breeding, or introducing alien pathogens [1]. Few studies on the introduction of alien parasites have been carried out for both terrestrial and aquatic animals. Most of them are reported in fishes, and only a few in mammals. In most cases, the impact of the introduced parasite is significantly higher on the native species than on the co-introduced alien host, raising important concerns about the sanitary threat represented by the biological invasions [3].

The Eastern cottontail *Sylvilagus floridanus* is a native American lagomorph, whose range extends from southern Manitoba and Quebec to Central and north-western South America. In the United States, the Eastern cottontail ranges from the east to the Great Plains in the west [4]. It is a species with the capacity to adapt to a variety of environmental conditions, as it is able to use diverse and fragmented habitats [5,6]. For this reason, after introduction in new areas it can become invasive and quickly expand its range [7], competing with other lagomorphs for space and resources [5]. *S. floridanus* was introduced for hunting in several European countries during the second half of the 20th century: France (1953), Italy (1966), Spain (1980), and Switzerland (1982) [8,9]. Hunting is an activity supported by the movement, translocation, and introduction of millions of animals. In particular, 24.3% of mammals and 30.2% of birds introduced into Europe during the last century were released for hunting [10].

The only population of *S. floridanus* that was permanently established in Europe is the Italian one [11]. Twelve individuals were released in Turin Province (Piedmont Region—Northwestern Italy) [12]. Following additional releases [8], the population quickly spread, and it is now present in North and Central Italy [13]. In Italy, the Eastern cottontail is well adapted to human-disturbed habitats, especially agricultural landscapes mixed with patches of natural vegetation [14]. In these areas, *S. floridanus* may act as a biological competitor of the native European brown hare *Lepus europaeus* [15]. Several studies have suggested there is competition for space and food between the cottontail and other lagomorphs [7,16,17,18], but hares and cottontails can coexist if the shared habitats present a high degree of heterogeneity [14,19]. Recent studies have highlighted the possibility of indirect competition by a disruption of predator–prey dynamics [20]. The massive abundance of cottontails causes an increase in predator (fox *Vulpes vulpes*) presence, with detrimental effects on hare populations. The introduction of *S. floridanus* in Italy is an example of a biologic invasion following the uncontrolled release of an alien species. Biological invasions are considered among the major threats to biodiversity [21,22]. Invasive species may introduce viruses, bacteria, and parasites to which native species are more susceptible [23,24,25,26,27]. Parasites are also able to adapt to new ecological situations through selection of genotypes particularly suitable for the new environment [28]. Considering competition with native species and the demonstrated role in introducing exotic pathogens, *S. floridanus* can be definitively considered an invasive species with negative effects at an ecosystem level. The introduction or elimination of a parasite in an ecosystem can strongly affect the interactions between a diverse range of species in the community, and hence affect biodiversity [29].

The gastrointestinal community of Eastern cottontails in its native range is very diverse with at least thirty different gastrointestinal parasites reported: 11 cestodes, one trematode, and 18 nematodes [30,31,32,33,34,35,36,37,38,39,40,41,42,43,44]. Among them, the most frequently reported are *Taenia pisiformis*, *Obeliscoides cuniculi, Dermatoxys veligera, Trichuris leporis, Trichostrongylus calcaratus,* and *Hasstilesia tricolor*. Species reported at high or low prevalence are reported in Table 1.

Since its introduction in Italy in 1966, few studies have been carried out on this species. Among them, some evaluate sanitary threats from the presence of this alien species including macro- [45,46,47,48,49,50,51] and microparasites [52,53]. These studies reported exotic parasites (*O. cuniculi*) in native lagomorphs (*Oryctolagus cuniculus* and *L. europaeus*), as well as some preliminary data on the parasite community in *S. floridanus*. A couple of studies deal with the role of *S. floridanus* in the dynamics of important viral diseases, such as European Brown Hare Syndrome and Rabbit Haemorragic disease [52,53].

While acknowledging previous work on the health status of *S. floridanus*, the lack of extensive sanitary monitoring and sampling of the Eastern cottontail motivated this study of the structure and epidemiological characteristics of the gastrointestinal parasite community in a high-density population of *S. floridanus* in Italy.

## 2. Materials and Methods

### 2.1. Study Area

The study was carried out in the Province of Alessandria (Piedmont Region—Italy), which covers an area of 25,383 km^2^, with 29% lowlands and 71% mountains and hills [54]. Three different zones (“Roleto”, “Tollara”, and “Sezzadio”), where there are high population densities of *S. floridanus* (30–40 animals/km^2^), were selected for parasite monitoring (Figure 1) [14]. The first introduction of *S. floridanus* in the Piedmont took place in 1966, in the municipality of Pinerolo (Piedmont, Northern Italy), where three males and nine females were released along the Pellice river. From there, the species slowly spread to the rest of the region, including the Province of Alessandria. In the 1980s, the Eastern cottontail was very localized in the Alessandria Province. The introduced ranges underwent a marked increase in 1996/1997 and expanded throughout most of the province in the following years [12]. The Eastern cottontail shares these zones with the European hare *L. europaeus*, whose density varies according to restocking programs and habitat quality, up to a maximum density of 50 hares/ha [55]. In recent years, a decline in the hare population has been observed (Province of Alessandria—unpublished data) while the Eastern cottontail increased its range and density [14,15]. This observation suggests a potential ongoing competition between the two species. The dynamics of the two species in the Alessandria Province is highlighted in Figure 2, which shows the number of *L. europeaus* and *S. floridanus* estimated in protected areas from 1998–2011, following the first observation of Eastern cottontails in this zone. The observed trend of the two species, even if they suggest some competition between European hares and Eastern cottontails, should be cautiously interpreted, as the European hare population was already decreasing before the introduction of the Eastern cottontail. A comparison with *L. europaeus* dynamics in areas without *S. floridanus* would be necessary to derive more robust conclusions (but is outside of the scope of the present work).

### 2.2. Sample Collection

The sample of *S. floridanus* included animals shot under a program for the control and eradication of the species (as per article 29 of the Piedmont Regional Law n. 70/96). Beginning in 1999, the Alessandria Province started a program that permitted extensive hunting and capture of Eastern cottontails to control and eventually eradicate the species from the territory. Authorized operators conducted the program and recorded all the killed animals in a field registry. This operation is ongoing. The animals examined for our study were collected from June 2016–May 2017. Sex and weight were registered, and animals were grouped into two age classes: younger and older than six months, according to body weight [56]. Although body weight is not a completely accurate indicator of age, it provides a rough indication of physical maturity [57]. A total of 271 animals were collected: 90 in Tollara (46 males/44 females); 90 in Roleto (45 males/45 females); and 91 in Sezzadio (41 males/50 females). The distribution of the sample by age is shown in Table 2. The average weight of animals in the three study areas was 1042.07 kg +/− 210.46 in Tollara, 1071.93 kg +/− 245.07 in Roleto, and 1052.33 kg +/− 241.28 in Sezzadio. The average weight of females was 1119.96 kg +/− 259.13 versus 987.84 kg +/− 177.69 for males.

### 2.3. Parasitological Analysis

Necropsy was carried out for all the animals. The stomach was opened along the greater curvature; small and large intestines were opened longitudinally. The contents were washed into individual jars and settled for 30 min. The sediment was shed into a Petri dish with a black background. The content was diluted in 1 L of water, and 10% of the total was examined. If no parasite was found, the rest of the content was inspected. Parasites were stored in Eppendorf tubes with 70% ethanol [58], observed with a light microscope (20× and 40× magnification), and identified according to Skriabin [59]. In particular, the shape and size of *spicula*, *gubernaculum*, and other reproductive structures of the nematodes were used to identify the parasite to the species level.

### 2.4. Epidemiological Analysis

Nematode prevalence (P—percentage of infected animals), intensity (I—mean number of parasites per infected animals), and abundance (A—mean number of parasites per examined host) were computed. Data were analyzed using R 3.5.0 software [60]. Descriptive statistics were used to characterize variations in the main epidemiological indices among the three study areas.

## 3. Results

Three nematode species were identified: *O. cuniculi* in the stomach, *T. calcaratus* in the small intestine, and *P. nonannulatus* in the large intestine. Their prevalence (P), intensity (I), and abundance (A) are reported in Table 3.

The parasite community structure was quite different in the three study areas. In Sezzadio, *O. cuniculi* showed lower values for both prevalence (P = 5%, compared to the 84% registered in Tollara and 93% in Roleto), intensity (average of 23.4 vs. 52.68 and 77.55 recorded in Tollara and Roleto, respectively), and abundance (average of 1.29 vs. 44.49 and 72.38 recorded in Tollara and Roleto, respectively) (Figure 3). *T. calcaratus* showed similar prevalence values among areas. Also in this case, Sezzadio was the zone with lowest abundance and intensity values. Finally, *P. nonannulatus* was found only in one sampling area (Roleto), at quite high prevalence, abundance, and intensity.

## 4. Discussion

This work was carried out to shed light on the parasite community of an invasive species outside its natural range. The sample size in this study was quite large compared to previously published studies on Eastern cottontail parasites [33,37,44], allowing accurate estimation of the main epidemiological indices of the parasite community. The sex ratio ratio and age distribution in our sample are within previously reported ranges [61]. The mean weight of the females was higher than that of the males, in line with previous studies [57,62,63,64]. These data allowed us to consider the sample collected as representative of a typical Eastern cottontail population.

The main result of the study is the finding of three non-native parasite species. The occurrence of these exotic species was noted in previous works [47,48,49,50,51] but without an extensive sanitary surveillance. Our work describes the main epidemiological indices of an Eastern cottontail parasite community outside its native range, highlighting some key ecological findings.

The parasite richness detected in our study was significantly lower than that reported for *S. floridanus* in its native range. We identified only three species out of the 30 gastrointestinal helminths reported in literature, corresponding to a 90% reduction in parasite community richness. Reductions of parasite richness in an introduced species may bring just a “selection” of the original parasite community [65]; moreover, the higher fitness of an introduced species can be explained by the enemy release hypothesis [66]. Specifically, in its native range, a species has co-evolved with several factors (pathogens, parasites, predators) that limit its population. When introduced into a new area, it leaves these limiting factors behind, improving its fitness. On the contrary, we did not find any parasite species typical of the native lagomorphs, suggesting either a strong adaptation of the Eastern cottontail to its parasites or a resistance to local infections.

*O. cuniculi* was present in all three zones with a prevalence similar to previous reported [33,44]. In particular, the prevalence in Roleto (93%) and Tollara (84%) is among the highest reported in the literature [38,39,44]. Lower prevalence, as in Sezzadio (5%), has been reported by Moore and Moore [33] in Alaska (0.58%), Alabama (8.8%), and Oklahoma (10.9%). Also, the intensity values in Roleto (77.55) and Tollara (52.68) were higher than those reported in the literature (maximum of 31 parasites per animal) [41,42,44].

*T. calcaratus* was found in all the study areas, also in this case with higher prevalence (Roleto 90%, Tollara 87%, Sezzadio 88%) than that reported in the native range (between 3.1% [34] and 85% [39]). Abundance was similar (172.73–251.91) to what has been described by Boggs et al. (117.7–405.9) [36], while intensities (Roleto 234.23, Tollara 286.99, Sezzadio 196.5) were higher than those reported in Kentucky (12 nematodes per infected animal) [41] and in Illinois (80 nematodes per infected animal) [44].

*P. nonannulatus* was found only in Roleto, also in this case with a prevalence (62%) higher than that reported by Moore and Moore (0.6%) [33] and Erickson (17.9%) [34]. The absence of this parasite in Tollara and Sezzadio might be due to either a different origin of the introduced *S. floridanus* population or an earlier extinction of this nematode. Roleto is a lowland zone while Tollara and Sezzadio are hilly areas, thus different environmental conditions might have limited the presence of this parasite. The three species found in this study are among the ones with the highest prevalence described for Eastern cottontail. The parasite species “left behind” are, in fact, those that are less prevalent in native populations [65]. In the Alessandria Province, parasite prevalence was among the highest ever reported in the Eastern cottontail. All the epidemiological indices of the three parasites clearly described a parasite community less diverse but with significantly higher prevalence, intensity, and abundance values than the ones reported in the native range. The loss of some species and the consequent reduction of interspecific competition at the parasite community level may have provided better conditions for these parasites to colonize the host more severely.

The parasite community structure was quite variable among the sampling areas. As previously stated, differences related to sampling areas may suggest that the introduced populations have distinct origins. Other factors include environmental conditions that could influence the development of the parasite’s infective larval stages [67,68].

The lack of natural enemies can favor the ecological success of introduced species [65]. When an invasive species leaves some parasites behind, release from those parasites can cause a demographic explosion leading to higher population densities and average body weight [69]. The reduced parasite richness observed in the introduced Eastern cottontail may contribute to the success of *S. floridanus* in colonizing new areas. Considering the above, the ecological impact of *S. floridanus* is of concern, as the release of this lagomorph caused the introduction of four alien species in a new ecosystem: one mammal and three nematodes. The presence of the parasite species was undetected for several decades, and their effects at an ecosystem level are unknown.

The risk of spillover of these parasites to native lagomorphs is quite high. This phenomenon has been reported for the European brown hare in the same area for some nematode species [47,49], and it could potentially cause parasite-mediated competition with unknown consequences for the ecological balance of native animal populations. On the other hand, no public health concern has been highlighted in our study, as none of the introduced nematodes is zoonotic. Further studies on *S. floridanus* sanitary status in Italy and on the parasite epidemiological characteristics in sympatric wild lagomorph populations are needed to better understand how this scenario may evolve and the possible long-term consequences on biodiversity conservation.

## Figures and Tables

**Figure 1 life-10-00142-f001:**
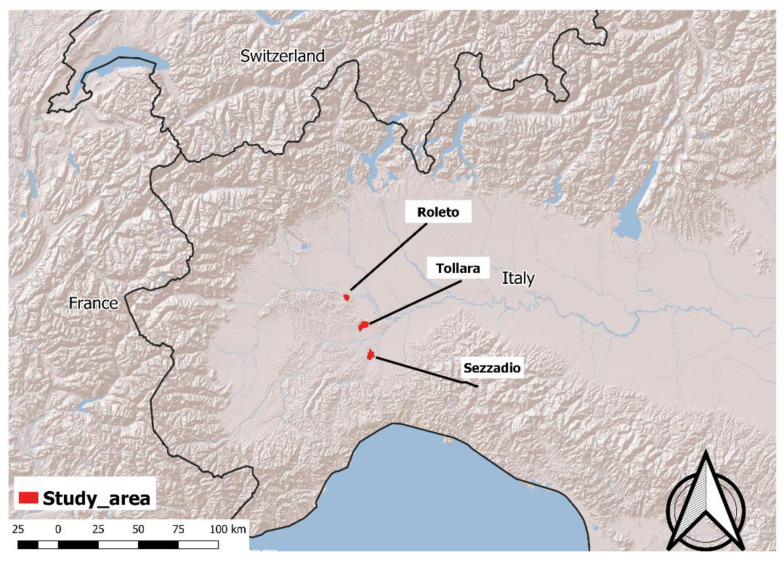
Alessandria Province (Italy), the three study areas (Roleto, Tollara, and Sezzadio) are marked in red.

**Figure 2 life-10-00142-f002:**
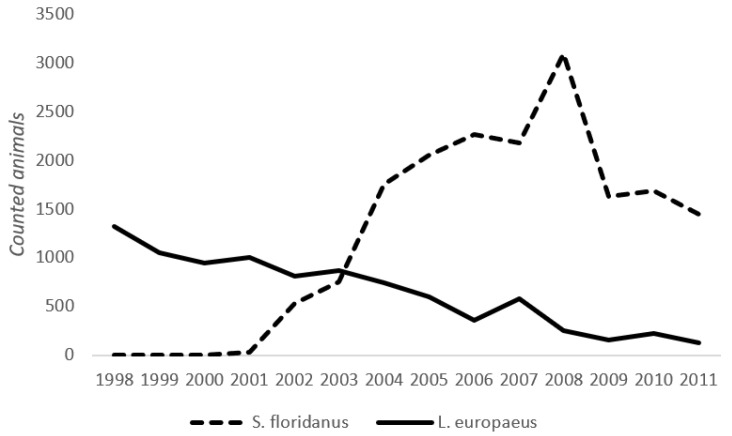
Dynamics of *L. europaeus* and *S. floridanus* in a protected area located in Alessandria Province during the period from 1998–2011.

**Figure 3 life-10-00142-f003:**
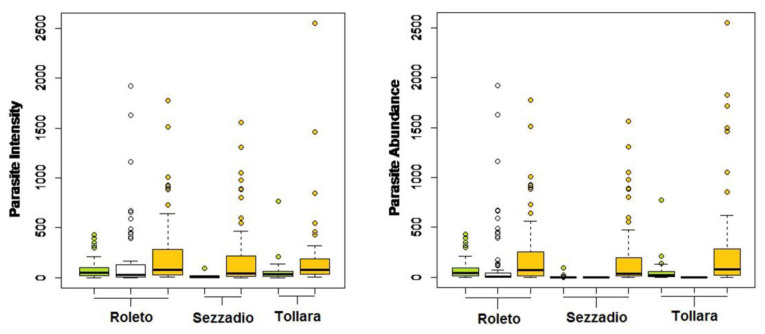
Intensity and abundance of the three parasites by study area (*O. cuniculi*—
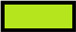
; *P. nonannulatus*—
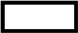
; *T. calcaratus*—
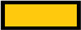
).

**Table 1 life-10-00142-t001:** A selection of the main parasite species described in *S. floridanus*, with high or low prevalence and maximum prevalence values reported.

Parasite Species	Prevalence Category	Maximum Prevalence Reported
*Trichostrongylus calcaratus*	High	85.0%
*Obeliscoides cuniculi*	High	96.0%
*Trichostrongylus affinis*	High	71.0%
*Longistriata noviberiae*	High	72.0%
*Nematodirus leporis*	Low	15.2%
*Passalurus nonanulatus*	Low	17.9%
*Trichuris sylvilagi*	Low	3.2%
*Physaloptera* sp.	Low	1.0%
*Gongylonema pulchrum*	Low	1.0%

**Table 2 life-10-00142-t002:** Eastern cottontails divided by age group and study area.

Age Class	Tollara	Roleto	Sezzadio
Younger than six months	52 (57.78%)	49 (54.44%)	52 (57.14%)
Older than six months	38 (42.22%)	41 (45.56%)	39 (42.86%)

**Table 3 life-10-00142-t003:** Prevalence % (P), mean intensity (I), mean abundance (A), standard deviations (SD), median (Med), minimum (m), and maximum (M) values of the parasite species.

Parasite	P	I (SD)	A (SD)	Med	m	M	Study Area
*O. cuniculi*	93	77.55 (91.57)	72.38 (90.54)	42.5	0	430	Roleto
	84	52.68 (92.39)	44.49 (86.96)	20	0	770	Tollara
	5	23.4 (38.04)	1.29 (9.65)	0	0	90	Sezzadio
*T. calcaratus*	90	234.23 (347.12)	210.81 (336.6)	72	0	1778	Roleto
	87	286.99 (463.95)	251.91 (444.5)	79.5	0	2550	Tollara
	88	196.5 (313.27)	172.73 (301.6)	32	0	1560	Sezzadio
*P. nonannulatus*	62	191.88 (387.76)	119.39 (318.86)	4	0	1920	Roleto

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
