# Peer review of "Does the Introduction of Alien Species Represent a Sanitary Threat for Native Species? The Case of the Eastern Cottontail Sylvilagus floridanus in Italy"

_life, 2020, doi:10.3390/life10080142_

Round 1
Reviewer 1 Report
The paper LIFE_800422 deals with the parasitology of introduced cottontails in Northern Italy. Despite being not a parasitologist, I really like this paper and I think that it could provide useful news in the field of biological invasions. Therefore, I strongly recommend it for publication after some minor revisions have been considered.
First, when I worked abroad, most researchers told me to avoid – when possible – to cite works written in languages different from English (which can be understood worldwide), such as German or France or Italian too, or non-peer-reviewed ones. These last can be used only when peer-reviewed papers are not available, but it could not be always the case. Therefore, I would like to ask you to improve your literature search.
ABSTRACT. I suggest you to change the first with the second sentence. Conversely, it is unclear who “It” at line 14 refers to.
LINE 15 and 47. What do you mean with “adapted”? Maybe coevolved? May you clarify? Maybe it is a correct term in parasitology, but I am not a parasitologist…
LINES 38-39 Instead of Angelici et al., which may be hard to be read for international authors, I suggest you to read something more recent (including the total Italian distribution of this species), such as: Dori P., Scalisi M., Mori E. (2019). “An American near Rome”… and not only! Presence of the eastern cottontail in Central Italy and potential impacts on the endemic and vulnerable Apennine hare. Mammalia 83: 307-312.
LINE 42. This is not a good reference for potential competition between cottontails and hares. I suggest you to read these papers: Vidus-Rosin A, Meriggi A, Cardarelli E, Serrano-Perez S, Mariani MC, Corradelli A, Barba A (2011) Habitat overlap between sympatric European hares (Lepus europaeus) and Eastern cottontails (Sylvilagus floridanus) in northern Italy. Acta Theriologica 56: 53-61. Vidus-Rosin A, Lizier L, Meriggi A, Serrano-Perez S (2012) Habitat selection and segregation by two sympatric lagomorphs: the case of European hares (Lepus europaeus) and Eastern cottontails (Sylvilagus floridanus) in northern Italy. Acta Theriologica 57: 295-304.
LINES 42-43. I suggest you to change “demonstrated” with “suggested”, given that strong evidences are needed to demonstrate competition. Furthermore, you also cited French papers where cottontails failed to establish, thus making it not a strong competitor. This should be clarified.
LINE 51. Are these parasites reported in the native or in the introduced range?
LINES 52-53. What does being reported in many papers mean? Are they more frequent or are they found rarely and thus deserving more attention (and more papers)? I do not think that number of papers can be considered as a synonym of infective importance.
LINE 69. When did you carry out the study? The paper you cited for reporting a local “high density” dates back to 2009 and - in 11 years - things may have changed.
METHODS AND RESULTS. “Roleto”, “Sezzadio” and “Tollara” may mean little to who is not from Piedmont. Maybe, it would be better to use something like “Area 1”, “Area 2” and “Area 3”.
LINE 98. Is “prevalence” the same of “percentage of infected animals”?
LINES 122-123. The paper you cited dates back to 2004. Please, search for something much more updated.
LINES 143-144. I suggest you to discuss this sentence also according to the “enemy-release hypothesis”.
REFERENCES: Check throughout the reference list, as I spotted many scientific names are not in italics. (e.g. refs 3, 38, 40, 41, 42)
Reviewer 2 Report
This study describes the gastro-intestinal helminth community of Eastern cottontails introduced in Italy. The ms is well written and clear, and offers an accurate description of invasive cottontails' parasite community, making use of a very good sample size. There were only a few reports and sparse data about parasites of introduced cottontails until now, and this ms fills this gap by adding a complete description and epidemiological analysis, and rising interesting questions about the origin of introduced populations and about potential parasite-mediated competition with native, declining lagomorph populations.
Main comments:
- The title suggests that you demonstrated that cottontails currently represent an actual sanitary risk, which you did not, as we do not know whether these alien parasites may infect humans, domestic animals or native wildlife and what impact they would have. I would keep it more general or use an interrogative form. Also, change "allochtonous" with "invasive" or "alien".
- The terms "authochtonous" and "exotic/allochtonous" should be changed throughout the text with terms "native" and "alien" (or "non-native"), which are more widely used in invasion ecology. Cottontails can be also referred to as "invasive".
- Sample collection: When were the animals sampled? Sampling season (and even year) may greatly affect parasite epidemiology
- Epidemiological analysis: Were the data truly normally distributed (Shapiro-wilk p > 0.05)? It would be surprising (especially for abundance) since helminths have usually an aggregated distribution within the host population.
- Epidemiological analysis: If I understood correctly, to analyse variation in abundance and intensity you first performed an ANOVA to test the effect of study areas and then two t-tests for sex and age class. Why not simply use a multifactorial ANOVA and consider all the three factors together? It would provide a more powerful analysis and you could test for interactions (area*age; area*sex and sex*age) as well, since your data look well balanced and you have a good sample size.
- Discussion: you state ( for example at lines 139-141 but in the abstract as well) that, except for O. cuniculi, this is the first report of the other parasites in Europe. However, in Tizzani et al. (Parasitol Res, 2011) some of you state at the beginning that T. calcaratus and P. nonannulatus had been reported in the cottontail in Italy. Additionally, T. calcaratus had been previously reported also in alien grey squirrels in Italy: Romeo C, Wauters LA, Ferrari N, et al (2014) Macroparasite Fauna of Alien Grey Squirrels (Sciurus carolinensis): Composition, Variability and Implications for Native Species. PLoS ONE 9:e88002.
Minor comments:
line 14: change "it" with "alien species"
line 14: why only mammals? this is true for all species
line 23: change to "introduction of non-native vertebrates"
line 32: change "an invaders" to "invasive"
line 33: change "areal" to "range"
line 40: change "above all" with "especially"
line 42: change "researches" to "studies"
line 42: change "with" to "between the cottontail and", otherwise the subject can be misunderstood
lines 48-49: it is not clear what you mean with this sentence, please rephrase
lines 50-58: all this part would be better and more clearly presented with a table. If there is a limit on the number of tables that can be included, I would perhaps remove Table 1 as that information can be easily added in the text.
line 98: change "infested" with "infected". In English to infest is normally used only for ectoparasites
line 99: change abundance definition to "mean number of parasites per examined host"
line 102: change the sentence to : "..to evaluate variation in parasite abundance and intensity by study area, sex and age"
line 129: change "only few" with "only a few"
line 132: add "community" after "parasite"
line 148: why should the lack of local parasites suggest a strong adaptation of the cottontail to its own parasites? This is independent from new infections. To me it would rather suggest a resistance to local infections at the most
line 173: change to "at the parasite community level"
line 174: change "guarantee" with "guaranteed"
line 175: change "Yet" to "Although"
line 178: change "was" with "were"
line 178-179: English need to be revised
line 180: "Sampling area"
line 181-182: change the sentence to "may suggest that the introduced populations have distinct origins"
line 182: change "areas diversity" with "differences among areas"
line 184: change "infesting" with "infective"
line 185: in this context, enemies include parasites, predators and grazers, so there is no need to specify predators
line 185: change "frequent reason provided" with "potential reason" and add a reference for this (e.g. Torchin et al., Nature, 2003)
line 186: delete "Moreover", you are expanding on the same concept you just mentioned
line 195: change "spill-over" to "spillover"
line 197: "parasite" singular
line 198: change to "..for the ecological balance of native animal populations"
Figure 1: remove the titles above the graphics and add instead parasite intensity/abundance - and optionally study area - as axis titles; change "according to" with "by" in the caption
Reviewer 3 Report
Dear authors,
I had the pleasure to review your study, entitled “Sanitary risk represented by the introduction of allochthonous species: the case of Eastern cottontail in Italy”.
The study itself is quite interesting and you have some data that must be published, because they cover the sanitary situation of invasive Eastern cottontail, which can be important for native lagomorphs in Italy.
However, the study has some major flaws, which do not make it suitable for publishing, without a major review, which should also include a change in how you analyzed your data. The most important points are:
I do not understand the overall rational of your study. If your best findings are about parasite diversity in cottontails in Italy, compared to parasite loads of cottontails in America, why not comparing this sample with data from the literature? There are many studies, published between 1950’s and 1990’s about parasites in cottontails, which might help you with comparisons;
I assume that some hypothesis of statistical testing were ‘silly nulls’: why are you comparing parasite loads between age classes, sexes and areas?
I would be cautious about the statistical power, as well as about using null hypothesis testing;
The introduction does not cover some relevant literature about invasive S.floridanus in Italy and does not explain well the scope of the study;
the study does not provide reproducible software code and individual data about animals, which can be fundamental for an evolving field such as invasion biology.
Some more comments:
Title – The title is misleading. The sanitary risk posed by invasive alien species goes well beyond the risk of parasite transmission to native lagomorphs from S.floridanus in Italy. You should be more specific. Apart from being misleading, such of a title does not appear on Google Scholar when someone searches about S.floridanus. Be more specific.
Lines 40 – 42: actually there is evidence for apparent competition, mediated by an increased risk of predation from the red fox. See: Cerri, J., Ferretti, M., & Bertolino, S. (2017). Rabbits killing hares: an invasive mammal modifies native predator–prey dynamics. Animal Conservation, 20(6), 511-519.
Lines 59-63: although the number of studies is not really high, they constitute a good proportion (I would say one third, maybe even more) of the research about S.floridanus in its invaded range. Moreover, you are ignoring some studies about viruses, such as:
Lavazza, A., Cavadini, P., Barbieri, I., Tizzani, P., Pinheiro, A., Abrantes, J., ... & Meneguz, P. G. (2015). Field and experimental data indicate that the eastern cottontail (Sylvilagus floridanus) is susceptible to infection with European brown hare syndrome (EBHS) virus and not with rabbit haemorrhagic disease (RHD) virus. Veterinary research, 46(1), 13.
D'Angelo, A., Cerri, J., Cavadini, P., Lavazza, A., Capucci, L., & Ferretti, M. (2019). The Eastern cottontail (Sylvilagus floridanus) in Tuscany (Central Italy): weak evidence for its role as a host of EBHSV and RHDV. Hystrix, the Italian Journal of Mammalogy, 30(1), 8-11.
Lines 71-74: “ In the last years, a decline in the hare population has been observed (Province of Alessandria – unpublished data) while the Eastern cottontail is progressively increasing its areal and density [9, 10]. This observation suggests a potential ongoing competition between the two species.”. There are a few things that must be rephrased:
I do not trust unpublished data, you should provide some evidence of what you are saying, also by adding some tables with some census estimates, or some data from hunting bags;
the two studies you cite have been carried out more than a decade ago, and therefore the cottontail could have declined as well since then. I mean, I do not believe that S.floridanus is collapsing, because the species is more widespread than ever in Northern Italy (see: Dori, P., Scalisi, M., & Mori, E. (2019). “An American near Rome”… and not only! Presence of the eastern cottontail in Central Italy and potential impacts on the endemic and vulnerable Apennine hare. Mammalia, 83(3), 307-312.), but you cannot argue about species-specific trends from absent data or from studies carried out 10 years ago;
opposing trends in two potentially competing species do not necessarily signal competition. They can also be related to temporal changes in environmental confounders affecting both (e.g. long term changes in landscape which provide more permanent vegetation benefiting cottontails and limiting hares, or climatic changes). Furthermore evidence about competition between the two species is less straightforward than the on you presented. Probably the most reasonable interaction is apparent competition, as specified above;
you are mentioning competition, but aren’t you studying parasites? In this case it would be more appropriate to talk about apparent competition, where two species compete through a mediation from a shared pathogen. There is a certain confusion between direct and indirect competition, in the introductory section;
you are mentioning 50 hares/hectare, from a management plan from 1999. You cannot use a document from 20 years ago as a reference for the density of a species like the hare, which is heavily culled and has considerable fluctuations in its populations. Moreover I am skeptical about this value, which seems unrealistic to me (50 hares in an area as large as one football pitch and a half) and maybe was obtained by counting hares on a spotlight surface only, without making inference about the whole area, after some restocking operation;
Introduction: you should state more clearly what the study aims are, at the end of the introduction.
Epidemiological analysis – There are various things that are not entirely convincing me:
you are analyzing with ANOVA, a procedure which is based on a Gaussian distribution of the observations, some data which should not be analyzed in this way. You cannot analyze percentages with an ANOVA, you should use some approach for proportions (e.g. a binomial generalized linear model, or a Beta regression: https://besjournals.onlinelibrary.wiley.com/doi/full/10.1111/2041-210X.13234). ANOVA in principle might be suitable for analyzing average parasite loads, but from your boxplots I notice that the distribution of your data is far from normality, being highly skewed.
You are using some tests, such as the Shapiro-Wilk test to assess the normality of your response variable. These tests are not the best approach for this task, they were widely adopted 20 years ago as computing them was easy. Nowadays, graphical exploration of the data is better, such as an histogram or comparing quantiles of the sample with quantiles from a Gaussian distribution (qqnorm and qqline functions in R) is better. Null hypothesis tests are quite sensitive to violations in the data, such as heteroskedasticity (see: Zuur, A., Ieno, E. N., & Smith, G. M. (2007). Analyzing ecological data. Springer.). Not surprisingly, from your boxplot, your data do not seem to follow a Gaussian distribution;
it is unclear what your hypotheses exactly are. If you are just comparing parasite loads between age classes literature already states that they should differ, because animals of different age are more or less sensitive to infestation. Also, explain why differences should occur between sexes or areas, by citing the relevant literature. Your approach seems to be based on some silly-nulls;
also when your data were collected? As parasite loads vary seasonally, it could be that an unbalanced data collection between areas (e.g. sampling one area in winter and one area in summer) could simply reflect seasonal fluctuations;
considered that your groups include 40-50 animals, I would also carry out a power analysis to see which differences should be detected, and at which power. I suspect that most slight differences (e.g. a parasite load difference smaller than 20%) cannot be detected with these numbers;
also, I would shift to a generalized linear model framework, which is more flexible, robust and clear than null hypothesis testing. See:
Cumming, G. (2014). The new statistics: Why and how. Psychological science, 25(1), 7-29.
Kruschke, J. K., & Liddell, T. M. (2018). The Bayesian New Statistics: Hypothesis testing, estimation, meta-analysis, and power analysis from a Bayesian perspective. Psychonomic Bulletin & Review, 25(1), 178-206.
please provide the R code and the dataset with individual information about culling;
Lines 122-127: this section is more suitable for the introduction. However, claiming that “few studies on the introduction of exotic parasites have been carried out” is false. There are at least dozens of studies about this, as introduced pathogens are a considerable impact of invasive species worldwide. It might be that these studies are a minority, in invasion biology, or that few of them focused on invasive mammals (I have some doubts about this point as well), but they are certainly not “few studies”.
Lines 135-140: interestingly, differences in sex-ratio or in sexual dimorphism could have been tested as well, by comparing Italian animals with data from the literature.
Line 143-144: I do not understand this sentence. You identified only 3 helminths, while 30 of them are reported for S. floridanus: does cottontails on average have 30 helminths inside their gastrointestinal trait, or are 30 of them reported? The two things are quite different.
Moreover: if you detected such of a striking difference, why not comparing your data with data from literature? You can actually compare the n. of helminths detected in your sample with a reference sample obtained from a meta-analysis. This would be more interesting, under an ecological viewpoint, than comparing groups of animals without any clear rationale.
Reviewer 4 Report
Title. The title should be modified, because some readers could interpret that the sanitary risk refers to human health. For instance: Sanitary risk to native lagomorphs represented by the introduction of allochtonous species: the case of Eastern cottontail in Italy
Line 15-17. They should indicate that the work was carried out in three zones of Italy.
Line 19-21. They should indicate the name of the two species whose presence in Europe has been reported for the first time in Europe.
Line 35. Please include this reference:
Delibes-Mateos M., Castro F., Piorno V., Ramírez E., Blanco-Aguiar J. A., Aparicio F., Mínguez L. E., Ferreira C. C., Rouco C., Ríos-Saldaña C. A., Recuerda P., Villafuerte R. (2018) First assessment of the potential introduction by hunters of eastern cottontail rabbits (Sylvilagus floridanus) in Spain. Wildlife Research 45, 571-577. https://doi.org/10.1071/WR17185
Line 44-47. The authors should clarify is the S. floridanus is considered an exotic species (no negative impacts) or an invasive species with negative impacts on the host ecosystems.
Line 46-49. In this part of the discussion, the authors should include some lines talking about the effect of allochthonous parasite species on autochthonous species.
Line 59-60. Since the authors provided 5 references about parasites of S. floridanus in Italy, they should include some lines talking about the main findings of these previous articles, and after that, they should specify the novelty of this study.
Figure 1. The size of the study areas in the map is too small, so I suggest enlarging the second frame showing only the three study areas.
Study area. Information concerning the history of the introduction of S. floridanus in the study area is missing.
Line 79-80. More information about the eradication program is necessary (e.g. season and years).
Line 96. The reference provided is very old, please replace it.
Line 101. ANOVA should be followed by a post hoc test. Please, indicate which post hoc test you performed, and explained it in the results.
Line 110-111. The results of the statistical tests are necessary.
Figure 2. In the figure of Parasite abundance, remove the category P. nonannulatus for Sezzadio and Tollara regions.
Line 118-119. The statistical results are missing. Although there are no significant differences, it is necessary include the results (the output of the statistical tests).
Line 122-131. This part of the discussion should be moved to the introduction. Include also the following reference about the role of hunting on the introduction of exotic species
Carpio, A.J., Guerrero-Casado, J., Barasona, J.A. et al. Hunting as a source of alien species: a European review. Biological Invasions 19, 1197–1211 (2017). https://doi.org/10.1007/s10530-016-1313-0
Line 180. Replace ‘Sapling areas’ by ‘sampling areas’
Line 182-184. Add a reference to support this sentence.
Reviewer 5 Report
The Msc deals with the nematode population structure and a few epidemiological concern within an introduced species, the Eastern cottontail.
As said by authors, only 3 of the 30 potentially species infecting the host were found.
I think, authors could have given some micropotographs of the nématodes and explained the criteria for the identification.
I think also that authors could have discussed if they think the species found could be zoonotic or not, instead of two pages of discussion only based of epidemiology, which, according to me, is far too much regarding the results.
To conclude, I think this datas could be published only in a short communication with a strong reduction of the discussion.
Reviewer 6 Report
When you mention a specific species the first time, you must write out the name in full, especially when considering that T. affinis could refer to either Trichiotinus affinis or Temnothorax affinis.
Due to the black and white print in Figure 2, it is impossible to distinguish between the O.cuniculi and T. calcaratus columns – both are light gray. A cross-hatching columns might be more suitable. In any case, the graph is confusing in its current state.
Round 2
Reviewer 1 Report
The MS has been reviewed and all of my previous comments addressed.
The MS now can be accepted for publication on LIFE
Author Response
Dear referee,
Thank you very much for your valuable suggestions that significantly improved the manuscript
Reviewer 3 Report
See my comments with the R code.

Reviewer 5 Report
I consider the revised manuscript suitable for publication in this new form as a short communication.
Author Response

(The authors gave the same response as above.)
